# Fine-Tuning of Piezo1 Expression and Activity Ensures Efficient Myoblast Fusion during Skeletal Myogenesis

**DOI:** 10.3390/cells11030393

**Published:** 2022-01-24

**Authors:** Huascar Pedro Ortuste Quiroga, Massimo Ganassi, Shingo Yokoyama, Kodai Nakamura, Tomohiro Yamashita, Daniel Raimbach, Arisa Hagiwara, Oscar Harrington, Jodie Breach-Teji, Atsushi Asakura, Yoshiro Suzuki, Makoto Tominaga, Peter S. Zammit, Katsumasa Goto

**Affiliations:** 1Department of Physiology, Graduate School of Health Sciences, Toyohashi SOZO University, Aichi, Toyohashi 440-0016, Japan; s1755103@sc.sozo.ac.jp (K.N.); s1855104@sc.sozo.ac.jp (T.Y.); s1955103@sc.sozo.ac.jp (A.H.); 2Randall Centre for Cell and Molecular Biophysics, King’s College London, London SE1 1UL, UK; massimo.ganassi@kcl.ac.uk (M.G.); peter.zammit@kcl.ac.uk (P.S.Z.); 3Laboratory of Physiology, School of Health Sciences, Toyohashi SOZO University, Aichi, Toyohashi 440-0016, Japan; s-yokoyama@sozo.ac.jp; 4Centre of Human and Aerospace Physiological Sciences, King’s College London, London SE1 1UL, UK; danraimbach@gmail.com (D.R.); oscarharrington15@hotmail.co.uk (O.H.); jodiejoyb@yahoo.co.uk (J.B.-T.); 5Paul & Sheila Wellstone Muscular Dystrophy Center, Department of Neurology, Stem Cell Institute, University of Minnesota Medical School, Minneapolis, MN 55455, USA; asakura@umn.edu; 6Division of Cell Signalling, National Institute for Physiological Sciences, Aichi, Okazaki 444-0864, Japan; yoshiros@iwate-med.ac.jp (Y.S.); tominaga@nips.ac.jp (M.T.)

**Keywords:** skeletal muscle, satellite cells, mechanosensation, Piezo1, Ca^2+^ channel, myoblast fusion, growth, myogenesis, Fam38A

## Abstract

Mechanical stimuli, such as stretch and resistance training, are essential in regulating the growth and functioning of skeletal muscles. However, the molecular mechanisms involved in sensing mechanical stress during muscle formation remain unclear. Here, we investigated the role of the mechanosensitive ion channel Piezo1 during myogenic progression of both fast and slow muscle satellite cells. We found that Piezo1 level increases during myogenic differentiation and direct manipulation of Piezo1 in muscle stem cells alters the myogenic progression. Indeed, Piezo1 knockdown suppresses myoblast fusion, leading to smaller myotubes. Such an event is accompanied by significant downregulation of the fusogenic protein *Myomaker*. In parallel, while Piezo1 knockdown also lowers Ca^2+^ influx in response to stretch, Piezo1 activation increases Ca^2+^ influx in response to stretch and enhances myoblasts fusion. These findings may help understand molecular defects present in some muscle diseases. Our study shows that Piezo1 is essential for terminal muscle differentiation acting on myoblast fusion, suggesting that Piezo1 deregulation may have implications in muscle aging and degenerative diseases, including muscular dystrophies and neuromuscular disorders.

## 1. Introduction

Skeletal muscles are highly specialised tissues composed of multi-nucleated, post-mitotic myofibres. Since nuclei within a myofibre (myonuclei) do not divide after development, the production of new myonuclei, to sustain muscle maturation and repair, is entrusted to muscle stem cells (named satellite cells (SCs)). SCs are found on the surface/periphery of postnatal skeletal myofibres [1,2,3] and in response to muscle damage or hypertrophic stimuli, rapidly activate to generate a myoblast progeny that proliferate, undergo myogenic differentiation, and finally fuse to repair damaged myofibres, resulting in regeneration of a functional muscle [4]. Therefore, alterations in the myoblast fusion machinery can have profound effects on regeneration efficiency and overall muscle function and health. An important process dictating myoblast fusion is mechanosensation, but how this is regulated in the muscle remains largely unknown.

Mechanosensitive (MS) ion channels are pore-forming membrane proteins that gate in response to mechanical stimuli applied on the cell membrane [5,6,7]. MS ion channels have been linked to many physiological processes associated with mechanosensory transduction, including osmoregulation, proprioception, hearing, touch, and blood flow regulation, to name a few examples [8,9,10]. Piezo1 and Piezo2 were first identified as the long-sought principal types of molecular force sensors (mechanosensors) in mammalian cells [5]. Piezo1, and similarly Piezo2, are very large proteins containing ~2500 amino acids with each subunit (three subunits per channel), having an estimated 24–40 transmembrane (TM) segments [6,8,11,12]. Characterisation of ionic selectivity revealed that Piezo1 was nonselective, permeating Na^+^, K^+^, Ca^2+^, and Mg^2+^ with a preference for Ca^2+^ [5,13]. Ca^2+^ regulation plays a crucial role in skeletal muscle maintenance and repair; thus, understanding Piezo1′s function may prove vital when developing therapeutic interventions for muscular dystrophies [14,15].

Here, we analyse the role of Piezo1 in skeletal myogenesis, focussing on muscle differentiation and its role in stretch-induced Ca^2+^ influx of primary myoblast-derived myotubes. Our findings reveal that Piezo1 is dispensable for myoblast proliferation and onset of differentiation, but it is finely regulated during myoblast fusion and myofibre maturation. Indeed, Piezo1 knockdown suppresses the formation and maturation of primary myotubes derived from mouse slow soleus (SOL) and fast extensor digitorum longus (EDL) muscles. At the molecular level, Piezo1 reduction leads to downregulation of the fusogenic gene *Myomaker*, decreases filamentous actin (f-actin) accumulation and organisation, and lowers Ca^2+^ influx of myotubes in response to mechanical stretch. In contrast, administration of the Piezo1-specific agonist Yoda1 increased myoblast fusion. Congruently, Piezo1 activation also showed increased Ca^2+^ influx in response to stretch. In summary, we show that Piezo1 plays a crucial role at the terminal stage of myoblast fusion and myofibre growth and maturation.

## 2. Materials and Methods

### 2.1. Primary Myoblast Cell Culture

All experimental procedures were carried out in accordance with the Guide for the Care and Use of Laboratory Animals as adopted and promulgated by the National Institutes of Health (Bethesda, MD, USA) and were approved by the Animal Use Committee of Toyohashi SOZO University (A2018006, A2019006). Male C57BL/6J mice (8–12 weeks of age) were used. All mice were housed in a vivarium room with 12-h–12-h light–dark cycle; with temperature and humidity maintained at ~23 °C and ~50%, respectively. Solid food and water were provided ad libitum. 

After cervical dislocation, the *extensor digitorum longus* (EDL) and SOL muscles were carefully dissected, and they were manipulated only by their tendons. Muscles were digested in 0.2% Collagenase Type 1 (Sigma, UK. Reference: SCR103) in Dulbecco’s Modified Eagle Medium (DMEM, Gibco, Thermo Fisher Scientific, Waltham, MA, USA, reference: 11885084) with 1% penicillin/streptomycin (Pen Strep, Gibco, Thermo Fisher Scientific, 15140-122) for 2 h. Individual myofibres were then dissociated by trituration using heat polished glass Pasteur pipettes (Marienfeld, Germany. Reference: 3233049) with variously sized apertures (coated with 5% BSA, Sigma-Aldrich, St. Louis, MO, USA, reference: A7906-100G) and washed as described by Collins and Zammit (2009) [16]. Isolated myofibres were plated on Matrigel (Corning, NY, USA, reference: 354234) and the SC-derived myoblasts were then expanded in proliferation medium, consisting of; DMEM, with 30% heat-inactivated foetal bovine serum (FBS) (Gibco, Thermo Fisher Scientific. Reference: 26140-079), 10% horse serum (Gibco, Thermo Fisher Scientific, reference: 16050-122), 1% chick embryo extract (Sera Laboratories., West Sussex, UK, reference: CE-650-TL), 10 ng/mL basic FGF (bFGF, Gibco, Thermo Fisher Scientific. Reference: PHG0264) and 1% penicillin. Cells were kept in a 37 °C incubator (Panasonic, Kadoma, Osaka, MCO-230AICUVH) under a humidified atmosphere with 95% air and 5% CO_2_. Cells designated for proliferation and differentiation conditions were seeded at different densities depending on the size of wells they were cultured in. The 96-well plate proliferation conditions required 5000 cells per well, and for differentiation, cells were seeded at 10,000 cells per well. For 6-well plates, proliferating and differentiation cohorts consisted of 50,000 cells and 70,000–80,000 per well, respectively. Differentiation medium was made up of DMEM, 2% heat-inactivated horse serum, and 1% penicillin.

### 2.2. siRNA Transfection

Small interfering RNAs (siRNAs) were purchased from (Qiagen, Hilden, Germany) (Table 1) and diluted to 20 or 10 µM in double-distilled water (ddH_2_O) and stored at −20 °C. To investigate the effects *Piezo1* knockdown on proliferation, early entry into differentiation and myotube formation, *Piezo1*-targeting or control scrambled siRNA (siScrambled; Qiagen, Hilden, Germany) was transfected in proliferation medium. Cells were plated on 6-well plates at 50,000 cells per well in proliferation medium. Following a 24 h incubation period, the medium was replaced with 1.75 mL fresh proliferation medium, and the transfection mixture was prepared: a solution 150 µL Opti-MEM (Gibco, Thermo Fisher Scientific. Reference: 31985-070) medium with 9 µL of lipofectamine (lipofectamine RNAiMAX Thermo Fisher Scientific. Reference: 13778030) was made for each well. Separately, siRNA was diluted in 150 µL Opti-MEM. The two solutions were then mixed and incubated for 5 min; 250 µL of the siRNA/lipofectamine mixture was added to corresponding wells dropwise. The final siRNA concentration was set at 10 nM. Following overnight incubation in the transfection medium, cells were trypsinised for RT-qPCR analysis and seeded in 96-well plates for proliferation and differentiation cohorts (day 1 and day 3 differentiation) in proliferation and differentiation medium, respectively. After a 24-h incubation, proliferating cells were subjected to a 2-h 5-ethynyl-2’-deoxyuridine (EdU) pulse and fixed as below. Day 1 and day 3 differentiating cohorts were also fixed.

To determine *Piezo1*’s role in myotube formation, siRNA transfection was performed in early differentiated myotubes. Cells were seeded at confluency in differentiation medium. Following 24-h incubation in differentiation medium, siRNA transfection was performed. Cohorts were designated for RT-qPCR analyses and immunolabelling. 

### 2.3. RNA Extraction and Reverse Transcription

RNA was extracted from cells using the RNeasy mini kit as per the manufacturer’s requirements (Qiagen. Reference: 217004). Reverse transcription was carried out using PrimeScript RT Master Mix (Takara Bio, Otsu, Japan. Reference: RR036A). Optical density analysis using a Nanodrop ND-1000 spectrophotometer (Labtech, UK) quantified RNA concentration. Samples were then loaded to a PCR thermal cycler (Takara, Dice mini). The resulting cDNA was then diluted 1:9 to obtain a working dilution for RT-qPCR analysis.

### 2.4. Real-Time Quantitative PCR (RT-qPCR)

Primers were designed using the Takara Bio Perfect Real Time Support System (Takara Bio, Table 2). Primers were diluted to 50 µM in ddH_2_O and stored at −20 °C. Real-time RT-qPCR was performed on the cDNA (Thermal Cycler Dice Real Time System IIMRQ, Takara Bio) using Takara SYBR Premix Ex Taq II (Takara Bio. Reference: RR802A). 12.5 µL of SYBR Premix Ex were added to each RT-qPCR well. 8.5 µL of ddH_2_O and 2 µL of the corresponding primers were then added (a final concentration of 2 µM per primer). 2 µL of the respective cDNA was then added to the appropriate wells, bringing the total volume to 25 µL per well. The RT-qPCR cycle consisted of 95 °C for 30 seconds (s) (for enzyme activation), followed by 40 cycles at 95 °C for 5 s and a qPCR amplification period of 30 s at 60 °C. The relative fold change of expression was calculated by the comparative threshold cycle (CT) method using Takara Thermal Cycler Dice Real Time System Software Ver. 4.00 (Takara Bio). To normalise for the amount of total RNA present in each reaction, *Gapdh* was used as an internal standard. 

### 2.5. Piezo1 Activation

In order to induce Piezo1 activation, early forming myotubes were subjected to the Piezo1 specific agonist Yoda1 (Cayman Chemical Company. Ann Arbor, MI, USA, reference: 21904) diluted in dimethyl sulfoxide (DMSO, Sigma. Reference: D2650-5 × 5 mL). This consisted of a 24-h incubation period in differentiation medium at high confluency (10,000 cells/well). By this point, myoblasts seeded at high confluency began to show myotube formation in the relatively small 96-well plates. These early formed myotubes were then administered Yoda1. Yoda1 binds the agonist transduction motif (ATM), located at the pore domain of the Piezo1 channel [17]. With each subunit displaying such motif, Yoda1 has potentially three binding sites. This phase of the investigation consisted of two main variables (1) drug concentration and (2) duration of time cells were incubated with the drug. Five concentrations were chosen in order to cover an increasing spectrum of Yoda1 final concentration, these were: 5, 10, 30, and 100 μM diluted in differentiation medium. Preliminary findings from the group found that a 24-h incubation with any of the concentrations chosen, led to complete abolishment of myotube maturation (data not shown). Thus, time-points thought to have potential to maximise myoblast fusion/myotube maturation were tested. The incubation time-points were set for 1 and 30 min, and 1 and 4 h. Control cohorts containing only DMSO were incubated at the allocated times to allow comparisons to be made within each condition. Following the incubation of Yoda1 or DMSO, cells were cultured in the differentiation medium for a further 2 days (i.e., myotubes were analysed 3 days post initial induction of differentiation).

### 2.6. Immunolabelling

Throughout the protocol, all washes were performed with Dulbecco’s phosphate-buffered saline (DPBS, Gibco, Dulbecco’s Phosphate Buffered Saline, Thermo Fisher Scientific, reference: 14190-144). Cells were fixed with 4% paraformaldehyde for 15 min. Samples were then washed three times with PBS (5 min each wash) and permeabilised for 15 min using 0.5% triton-X100/PBS (Sigma-Aldrich. Reference: T9284-500 mL). Cells were blocked for 1 h in 5% bovine serum albumin (BSA, Sigma-Aldrich reference: A7906-100 G). Primary antibodies (diluted to the working concentration in PBS) (Table 3) were added to the samples and incubated overnight at 4 °C. Primary antibodies were decanted, the samples were washed three times and appropriate secondary antibodies diluted to the working concentration in PBS added (Table 3) were added to the samples. The samples were covered with aluminium foil to avoid light exposure and left to stand at room temperature for 1 h. Cells were washed again (three times). To visualise nuclei, the cells were incubated for 10 min at room temperature with 1 µg/mL 4’, 6-diamidino-2-phenylindole (DAPI) (Sigma. Reference: D9542-10 MG) diluted 1:1000 in PBS. After a final wash with PBS (5 min), cells were replenished with PBS and stored at 4 °C until image analysis.

### 2.7. EdU Incorporation

For the evaluation of cell proliferation, cells were incubated with 5-ethynyl-2′-deoxyuridine (EdU: Invitrogen, Thermo Fisher Scientific) at 10 μM, added in fresh proliferation medium for 2 h at 37 °C. EdU, the alkyne-containing thymidine analogue, is incorporated into DNA during active DNA synthesis. The click-iT EdU Alexa Flour kit (Invitrogen, Thermo Fisher Scientific, Click-iT, EdU Alexa Fluor. Reference: 594 C10339) was used as per manufacturer’s instructions with either the 488 (green) or 594 (red) azide to detect incorporated EdU.

### 2.8. Phalloidin Labelling

To evaluate the cytoskeleton (f-actin), cells were treated with phalloidin (Invitrogen, Thermo Fisher Scientific, Alexa-Fluor. Reference: 488 A12379) diluted 1:40 in PBS. Phalloidin binds to f-actin, a major cytoskeleton protein in myofibres. Cells were incubated with phalloidin solution for 30 min at room temperature. Cells were then washed with PBS twice for 5 min.

### 2.9. Image Analysis and Quantification

Images were taken using a fluorescence microscope (BZ-X710, KEYENCE, Osaka, Japan). Moreover, 4 to five 5 images per each well (3 wells per repeat), consistent of a total of 12 to 15 images per repeat, were analysed. One repeat refers to one mouse. For EdU incorporation, the total number of DAPI-counterstained nuclei and total number of EdU-incorporated cells were quantified. The proportion of EdU-incorporated cells relative the total number of nuclei was subsequently expressed as percentages. The relative proportion of cells expressing Myogenin was also quantified in this manner.

The fusion index was calculated by quantifying the total number of nuclei within MyHC-positive myotubes (myonuclei) and expressing this value as a proportion of the total number of nuclei in each field of view. As a criterion, more than two nuclei must be within a MyHC-positive myotube to be quantified for the fusion index: (MyHC-positive myotubes containing ≥ 2 myonuclei/total number of nuclei) × 100.

To measure myotube width (diameter), the “measure” tool on ImageJ imaging software was used. This allows measurements of a chosen distance to be made. Before measurement, a scale was applied to all images. On the “set scale” option, pixels are converted into μm. Taking the fluorescence microscope and magnification into account, the program determines 100 μm to be 133.00 pixels or 0.75 μm/pixel. Three independent images were chosen per condition (nine pictures in total). The criterion for analysis was to choose the widest possible distance between myotube edges (i.e., myotube diameter) without the presence of any branching points. The values were then averaged. The myotube width distribution data (Appendix A) was quantified by measuring the diameter of all fibres in the field of view from 3 independent images from each mouse (nine in total). To plot distribution, these values were then divided into incremental bins of 5 µm and expressed as a percentage relative to the total myotubes per field of view (between 30 and 50 for controls and 50–100 myotubes for Yoda1 cohorts on average). Thus, a total of 90–150 control (DMSO) myotubes and 150–300 Yoda1-treated cohorts were analysed. The top five widest myotube diameters from each replicate (mouse) and condition (DMSO or Yoda1; fifteen measurements per condition) were taken and represented.

### 2.10. Stretch Experiments and Imaging of Intracellular Ca^2+^ Level

Stretch experiments were performed at the National Institute for Physiological Sciences (NIPS). Myoblasts were seeded (30,000 cells/chamber) on modified elastic silicone chambers (Strexcell, Ooyodonaka, Reference; STB-CH-0.02). After 24-h incubation, cells were transfected with either control (siScrambled) or *Piezo1*-specific siRNA in proliferation medium. After overnight incubation, cells were switched to differentiation medium and cultured for a further 3 days. Stretch experiments were conducted on the third day. For Piezo1 activation by Yoda1 administration, cells were seeded as above. Following the initial overnight incubation, cells were switched to differentiate and the resulting myotubes were analysed 3 days post differentiation induction.

For Ca^2+^ imaging, Fura 2-AM (Invitrogen) with 10% Pluronic^®^ F-127 (Molecular Probes, Eugene, OR, USA) diluted in double distilled water (ddH_2_O), was administered to EDL- and SOL-derived myotubes followed by a 30-min incubation time. Chambers were attached to an extension device (modified version of STB-150, Strex) on the microscope stage. Stretch stimulation was applied using a pre-set stretch speed and distance. After an initial 1 min rest period (0% stretch), stretch was applied at 3% (0.3 mm), 6% (0.6 mm), and 9% (0.9 mm) for 1 min, followed by a 1-min resting period in between. During the initial 0% stretch timepoint, Yoda1 cohorts were administered with 30 µM of the agonist before being subjected to stretch. Ionomycin (Sigma-Aldrich) at 5 µM was applied at the final step in each experiment for normalisation and to check cell viability.

Changes in intracellular calcium [Ca^2+^]i were measured by ratiometric imaging with Fura 2-AM at 340 and 380 nm, and the emitted light signal was read at 510 nm. Images were then analysed on ImageJ imaging software. Three independent myotubes from each condition were selected and analysed. The changes in ratios were calculated by subtracting basal values from peak values. The values were then normalised to ionomycin data.

### 2.11. Immunoblotting

Total protein lysates were obtained from EDL- and SOL-derived myoblasts and myotubes. After exposure to RIPA buffer (Thermo Fisher Scientific, Yokohama, Japan). Primary antibodies (Piezo1, Proteintech, 15939-1-AP) and β-tubulin (Wako, 10G10) diluted 1:100 and 1:5000 respectively in 5% skim milk, were incubated either for 90 min (β-tubulin) or at 4 °C overnight (Piezo1). Samples were then incubated with HRP-conjugated secondary antibodies (Cell signalling, anti-rabbit, 7074P2, and anti-mouse, 7076S) in 5% skimmed milk at 1:5000 dilution for 1 h at room temperature. Samples were visualised by chemiluminescence with a digital luminescent image analyser LAS-4000 (GE Healthcare, Tokyo, Japan).

### 2.12. Statistical Analysis

Data are presented as mean ± SEM from at least three experiments (at least three mice). Significance was assessed by either paired Student’s *t*-test or one-way ANOVA followed by followed by the Tukey-Kramer post-hoc; wherein *p*-values of <0.05 were considered to be statistically significant. A paired *t*-test was adopted when comparing effects within the same group e.g., analysing the effects of siRNA mediated down-regulation of *Piezo1* versus siRNA controls in murine derived myoblasts. A one-way ANOVA was implemented when two or more independent groups were analysed, for example, comparing the effects of varying agonist concentrations across different timepoints.

## 3. Results

### 3.1. Piezo1 Is Upregulated during Myoblast Differentiation

To investigate the expression level of *Piezo1* during myogenic progression, we used murine fast EDL and slow SOL muscle satellite cell (SC)-derived primary myoblasts (Figure 1a). The expression of EDL muscle SC-derived primary myoblasts showed a significant increase in mRNA expression of *Piezo1* in myotubes cultured at 3 days of differentiation, compared to the expression level in proliferating myoblasts (Figure 1b). In SOL-derived myoblasts, *Piezo1* expression rapidly increased after 24 h, suggesting an earlier function of Piezo1 in slow muscle myogenesis. Indeed, *Piezo1* was found upregulated at both 24 h (four-fold) and 72 h of differentiation in SOL-derived myoblasts compared to EDL-derived myoblasts (Figure 1b) where *Piezo1* increased by two-fold. Thus, *Piezo1* expression increases during myoblast differentiation. At the protein level, however, we find that both EDL and SOL-derived myotubes upregulate PIEZO1 comparably at the same level (Figure 1c–e). This may be due to transcriptional and translation differences between these muscle groups. Our results demanded we look further at the active role of Piezo1 in both EDL- and SOL-derived cells.

### 3.2. Piezo1 Regulates Ca^2+^ Influx during Myogenesis

The Piezo1 channel permeates Ca^2+^ influx at a greater preference than other cations (Na^+^, K^+^ and Mg^2+^) [5]. Ca^2+^ is itself a crucial regulatory of muscle contraction and earlier during muscle formation and differentiation/fusion [18,19,20]. Thus, given accumulation of *Piezo1* mRNA in differentiating myoblasts, we sought to assess the dynamics of Ca^2+^ influx ([Ca^2+^]i) upon modulation of Piezo1 activity in cultured myotubes. Using the customised stretch silicon bio-chambers [21], we cultured myotubes derived from both EDL- and SOL-derived myoblasts. We then divided the samples into two groups; those administered *Piezo1* specific siRNA (*Piezo1*-knockdown) and those given the Piezo1 agonist Yoda1 (at 30 µM). Results were compared against their respective controls. The chambers were subjected to incremental bouts of stretch, with a minute rest in between each stretch. Throughout the experiment we measured [Ca^2+^]i (Figure 2). *Piezo1* siRNA-mediated knockdown led to nearly 50% and 40% reduction on *Piezo1* mRNA in EDL-derived myoblasts (Appendix A) and SOL-derived myoblasts (Appendix A) respectively, compared to control conditions. Piezo1 knockdown was further confirmed through immunostaining (Appendix A). Under control conditions (siRNA controls and no Yoda1), upon mechanical stretch, [Ca^2+^]i increases significantly in both EDL-derived and SOL-derived myotubes compared to no-stretch (0%) controls (Figure 2a–h). Of note, for EDL-derived control myotubes only stretch bouts over 3%, showed a significant increase in [Ca^2+^]i. Reduction of *Piezo1* severely suppressed in [Ca^2+^]i in response to stretch compared to control cells (siRNA controls) (Figure 2a–d). Indeed, the progressive increase in [Ca^2+^]i was completely abolished in EDL-derived myotubes (Figure 2c) with neither a 6%, nor a 9% stretch eliciting a significant increase in [Ca^2+^]i (Figure 2c). Similarly, reduction of *Piezo1* in SOL-derived myotubes (Figure 2d), showed a significant decrease in [Ca^2+^]i at all stretch bouts, with only *Piezo1*-knockdown myotubes showing increased [Ca^2+^]i at the 9% stretch condition compared to 3% stretch Piezo1 siRNA cohorts. Thus, Piezo1 is essential for Ca^2+^ influx during stretch.

Yoda1-mediated activation of Piezo1 was able to lower its activation threshold. Indeed, we find that prior to stretch (0% stretch), Yoda1 administered myotubes already begin to show an enhanced [Ca^2+^]i compared to unstretched control counterparts (Figure 2e–h). Moreover, at bouts of 3% stretch, EDL- and SOL-derived myotubes treated with Yoda1 (Figure 2g,h) showed a significantly higher [Ca^2+^]i, compared to untreated 3% stretch counterparts. This increase in [Ca^2+^]i persisted at higher stretch bouts, with both EDL- and SOL-derived myotubes exhibiting higher [Ca^2+^]i at 6% and 9% stretch bouts compared to untreated stretched counterparts (Figure 2g,h). In summary, Piezo1 expression and activity are crucial for Ca^2+^ regulation in muscle function.

### 3.3. Piezo1 Is Dispensable for Myoblast Proliferation and Early Commitment to Differentiation

Piezo1 expression peaks at later myogenic steps, where it regulates calcium influx during contraction. However, whether Piezo1 participates in earlier phases of myogenesis is unclear. Thus, we set out to evaluate the effects of manipulating Piezo1 on proliferation and onset of myogenic program. Knockdown of Piezo1 had no overt effect on the proliferation rate of both EDL- and SOL-derived myoblasts (Appendix A), therefore Piezo1 is dispensable for muscle cell proliferation. Next, we investigated whether reduction of Piezo1 could alter the entrance into differentiation stage by analysing the accumulation of the transcription factor Myogenin. Neither EDL-derived or SOL-derived myoblasts showed a significant difference in the relative proportion of Myogenin-positive cells between *Piezo1*-knockdown and control-siRNA treated conditions (Figure 3a–d), suggesting that Piezo1 does not participate in the onset of myoblast differentiation. Piezo1 seemed to exert most of its expression during differentiation, thus in order to address potential compensatory effect of Piezo2 in response to Piezo1 knockdown, we measured *Piezo2*’s expression in EDL- and SOL-derived myotubes (Appendix A). We again confirmed the reliability in our method of siRNA-mediated knockdown by showing significant reduction of *Piezo1* expression in both EDL- and SOL-derived myotubes (Appendix A). Importantly, RT-qPCR analysis revealed that *Piezo1* knockdown does not alter *Piezo2* expression in EDL- and SOL-derived myotubes (Appendix A) indicating that Piezo2 does not compensate for Piezo1 suppression. Moreover, *Piezo2* expression was extremely low in both EDL and SOL samples, confirming previously published data [5,22].

Myoblast fusion requires extensive membrane remodelling together with mechanical stress; thus, we next evaluated the effect of Piezo1 suppression on myotube formation and maturation. Knockdown of Piezo1 led to a dramatic reduction in myoblast fusion and subsequent hinderance in the formation of both EDL- and SOL-derived myotubes (Figure 3e–h). In line with reduced myoblast fusion, EDL- and SOL-derived myoblasts (Figure 3i–l) showed reduced expression of the fusogen *Myomaker* [23] (Figure 3i,k) suggesting alteration of fusion machinery at the molecular level. Expression of *Myomixer* on the other hand, showed a trend to decrease (Figure 3j,l). Moreover, knockdown of *Piezo1* in early formed myotubes confirmed a significant reduction in the fusion index, compared to control-siRNA conditions (Figure 3m–p). This response persisted when using different Piezo1 targeted siRNAs on EDL- and SOL-derived myotubes (Appendix A). In summary, dysregulation of Piezo1 expression reduces the ability of cells to fuse into new or existing myotubes.

### 3.4. Piezo1 Activity Is Finely Regulated during Myoblast Fusion

Knockdown of Piezo1 reduces myoblast fusion and alters Ca^2+^ influx. We next assessed the dose-dependent effect of Piezo1 activation. Early forming myotubes were subjected to varying concentrations of Yoda1 over a period of time. After each allocated timepoint, the agonist containing medium was removed and replenished with fresh differentiation medium and incubated for a further two days (Figure 4).

Strikingly, a 1-min treatment of Yoda1 significantly enhanced cell fusion. both in EDL- and SOL-derived myotubes at 30 and 100 µM of Yoda1 (Figure 4a–d). However, a 30-min treatment with the highest dose (100 µM) of Yoda1 had the opposite effect, reducing fusion index (Appendix A), suggesting that Piezo1 activity must be finely tuned to achieve efficient fusion and myotube maturation. Indeed, both EDL- and SOL-derived myotubes incubated at 100 µM beyond the 1 min timepoint, showed a significant decrease in fusion, compared to vehicle-treated controls (Appendix A). At 30 min of incubation, both EDL- and SOL-derived myotubes showed increased fusion at 30 µM Yoda1 (Figure 4b,d). After 1 h of incubation timepoint, EDL-derived myotubes exhibited increased fusion efficiency at 5, 10, and 30 µM of Yoda1 treatment, compared to DMSO controls (Figure 4b). Similarly, SOL-derived myotubes incubated with Yoda1 for 1 h showed an increase in fusion at 5 µM, 10 µM and 30 µM, compared to DMSO controls (Figure 4d and Appendix A). Continued Piezo1 activation (4 h) showed a significant decrease in fusion of EDL-derived myotubes at a Yoda1 concentration of 30 µM compared to DMSO control (Appendix A).

At the molecular level, *Piezo1* expression was increased in both EDL- and SOL-derived myotubes post Yoda1 administration (Appendix A). *Myomaker* expression also paralleled *Piezo1* upregulation in both EDL- and SOL-derived myotubes (30 µM 30 min incubation, and 10 µM 1 h incubation, Appendix A). Interestingly, *Myomixer*, although showed a trending increase in expression, it did not reach statistical significance (Appendix A). Taken together, our data show that fine regulation of Piezo1 activity is a key step during differentiation dynamics.

We noticed that induced activation of Piezo1 appeared to affect myotube size, although increasing myoblast fusion. In order to address this, we compared the myotube width (diameter) of these samples to DMSO controls (Figure 4e–h). Interestingly, both EDL- (Figure 4e,f) and SOL-derived myotubes (Figure 4g,h) showed reduced myotube width compared to controls when treated with Yoda1. Both EDL- and SOL-derived myotubes, showed that Yoda1 treated cells, on average, have a greater proportion of smaller myotubes (Figure 4i,j) compared to controls, which have a higher distribution of lager myotubes. A similar pattern was observed in the rest of the Yoda1 treated cohorts that displayed increased fusion (Appendix A). Taken together, over-activation of Piezo1 unbalances myoblast fusion and myotube growth.

### 3.5. Deregulation of Piezo1 Alters Cytoskeletal Remodelling during Differentiation

Along with myoblast fusion, myogenic differentiation requires extensive cellular remodelling, and previous research places Piezo1 regulation as a key player in cytoskeletal homeostasis [21,24]. To understand whether Piezo1 contributes to regulate cytoskeletal structures, including f-actin, we examined f-actin accumulation as a proxy to evaluate the extent of cytoskeletal reorganisation during myogenic differentiation in EDL- and SOL-derived myotubes. *Piezo1*-knockdown showed a significant decrease in the accumulation of f-actin compared to control-siRNA (Appendix A), suggesting that reduction of Piezo1 alters cytoskeletal dynamics, as previously observed [21,25]. In contrast, f-actin accumulation was unaffected by Piezo1 over-activation (Appendix A), suggesting that Piezo1 may not directly contribute to f-actin remodelling. In line with alteration in myoblast fusion, excessive chemical activation of Piezo1 showed a significant decrease in f-actin in EDL (100 µM; 4-h incubation) and SOL (100 µM; 30-min, 1-h and 4-h incubation) -derived myotubes (Appendix A). These findings suggest that deregulation of *Piezo1* expression and/or its activation status impinge on cytoskeletal organisation during myoblast fusion and muscle differentiation.

## 4. Discussion

Our study reveals that the mechanosensitive ion channel Piezo1 is finely tuned during the myoblast fusion and formation of myotubes through four main findings. (1) Piezo1 expression increases during myoblast differentiation in a muscle-type independent fashion. (2) Piezo1 is essential for proper calcium influx during muscle contraction. (3) Modulation of Piezo1 expression alters both myoblast fusion and expression of the fusogens Myomaker. (4) Overactivation of Piezo1 disbalances myoblast fusion and myotube width.

### 4.1. Piezo1 in Myogenesis

The current study investigated the effects of Piezo1 regulation throughout the myogenic program. Piezo1 is expressed at a higher proportion in terminally differentiated myotubes, compared to proliferating myoblasts. Moreover, we found that, differentiating myocytes derived from the mainly slow-type muscle SOL displayed higher expression of *Piezo1* (at the mRNA level) compared to the fast EDL muscle. Understanding the potential differences in muscle/fibre types and Piezo1 regulation is an intriguing area for future research and could reflect differences in the dynamics of myogenic progression. We also confirmed that the expression of *Piezo2* is not altered by the downregulation of *Piezo1*. This is perhaps not surprising given the fact that *Piezo2* is not as abundant in skeletal muscle compared to *Piezo1* [5,22]. Nevertheless, it was important to see any potential compensatory effects Piezo2 may impose. We still, however, do not discount the possibility that Piezo2 may play a role in other areas of myogenesis, perhaps in myofibres or fully formed muscle groups. We must also acknowledge that perhaps if the degree of Piezo1 downregulation was more “potent”, Piezo2 may alter its expression in response. Furthermore, a prolonged downregulation or activation of Piezo1 may also affect Piezo2’s expression.

Specific downregulation of Piezo1 by siRNA-mediated transfection showed no significant change in the proliferation rate of either EDL- or SOL-derived myoblasts. However, our data do not exclude the possibility that Piezo1 is not involved in earlier myogenic events, perhaps in balancing quiescence and activation of SCs. In proliferating myoblasts, reduction of Piezo1 function does not alter onset of myoblast differentiation, evaluated by the proportion of Myogenin-positive cells. However, that is not to say that perhaps Piezo1 does not alter other events important to myoblasts such as cell motility, which the current study did not include. In our current study, where a significant phenotype was observed was in terminally differentiated myotubes. Indeed, our data found that knockdown of Piezo1 significantly reduced fusion of myocytes and prevented myotube formation and maturation. In contrast, activation of this Ca^2+^ permeable channel resulted in enhanced myoblast fusion.

Our finding shows that modulation of Piezo1 could alter cytoskeletal organization, in line with similar observations from other studies [21,25,26,27,28], indicating that Piezo1 cytoskeleton interplay could be conserved among cell types/tissues. However, although we could reliably measure the intensity changes of the cytoskeletal protein, the exact structural changes at the molecular level remain to be fully characterised.

### 4.2. Piezo1 Activation and Ca^2+^ Influx

We show that selective downregulation of Piezo1 dramatically suppressed [Ca^2+^]i, which most likely translates in the depression of the influx of Ca^2+^ into cultured myotubes exposed to stretch. In contrast, activation of Piezo1 significantly increased [Ca^2+^]i, which means the enhancement of Ca^2+^ influx. Our results propose that Piezo1 is a novel intracellular Ca^2+^ regulatory protein in skeletal muscle function. Ca^2+^ plays a crucial role in skeletal muscle function, maintenance, and plasticity. All myofibres use Ca^2+^ as their main regulatory and signalling molecule [18,19,20]. Therefore, the contractile functionalities of myofibres are dependent on the highly regulated expression of proteins involved in Ca^2+^ handling and signalling. Our study showed that Piezo1 mediated regulation of Ca^2+^ influx is a key driving factor in the respective decrease and increase in myoblast fusion in response to Piezo1 inhibition and activation. To the best of our knowledge this is the first time this has been demonstrated. An area of future research is to elucidate in detail the activation threshold of Piezo1. By increasing stretch conditions beyond 9% (e.g., 12–15%) and/or increasing Yoda1 concentration, we may determine the point at which Piezo1 is maximally activated. Moreover, future studies should explore the potential phenomena of inactivation or deactivation of the Piezo1 channel, in response to prolonged or repeated stimuli of skeletal muscle [5,13,24,29,30].

The silicon bio-chamber experiments revealed that the mechanical activation threshold of Piezo1 was severely blunted by Piezo1 downregulation. In a similar set of experiments (albeit using urothelial cells), Miyamoto et al. also showed a distance dependent increase of Ca^2+^ influx. Interestingly this response was blunted in Piezo1-siRNA-treated conditions [22]. The researchers also showed that a high enough [Ca^2+^]i must be attained in order to elicit a response, in their case ATP efflux. Our data support the presence of a stretch-dependent increase in Ca^2+^ influx. Remarkably we found that activation of Piezo1 resulted in increased [Ca^2+^]i even without stretch, suggesting that the activation threshold of Piezo1 was lowered. Furthermore, the data showed that reduction of Piezo1 expression significantly blunted any significant increase of [Ca^2+^]i in response to stretch. These results, for the first time show the need for Piezo1 to respond to stretch and permeate Ca^2+^ into myotubes. The findings also propose the presence of a physical threshold that must be attained before Piezo1 mediated Ca^2+^ influx is significantly increased. Moreover, as with Miyamoto et al. [22], we find that a stretch-dependent increase in Ca^2+^ influx is suppressed when Piezo1 expression is decreased. Conversely, we see an increase of [Ca^2+^]i when Piezo1 is activated. Whether this leads to altered cellular/myotube viability in the form of ATP release remains a subject for future research.

### 4.3. Piezo1 and Muscular Dystrophies

Piezo1 activation showed a significant increase in the fusion index of both EDL- and SOL-derived myotubes. Although this phenotype could be viewed as beneficial in terms of muscular dystrophy prevention, we must be aware of the potential dangers of an overactive Piezo1 channel. In fact, we showed that even a 30-min incubation of myotubes with a high agonist concentration (100 µM of Yoda1 treatment) led to decreased fusion in both EDL- and SOL-derived myotubes (Appendix A). This adverse phenotype is most likely the result of a dangerously high Ca^2+^ influx. Indeed, since the early 1990s, it has been postulated that defective mechanotransducers, also referred to stretch activated channels (SACs), contribute to the high influx of Ca^2+^ and, hence, maintain higher Ca^2+^ resting levels in Duchenne’s muscular dystrophy (DMD) [14,31]. However, the identity of this channel(s) in DMD is yet to be identified. The current data makes the hypothesis that DMD stricken muscle may have malfunctioning Piezo1 channels which permeate Ca^2+^ at hazardously high levels. It is possible that the reduced fusion we observed from an over-active Piezo1 channel (Figure 4 and Appendix A) stems from one of the many defects caused by over-accumulation of Ca^2+^. It is imperative that future experiments characterise the expression of Piezo1 not only in DMD but other muscular dystrophies.

### 4.4. Piezo1 and the Myoblast Fusion Machinery

Piezo1 downregulation significantly reduced myoblast fusion during myotube formation and myotube maturation. To the best of our knowledge there is only one other paper published that examined Piezo1 in skeletal muscle by Tsuchiya et al. [32]. Interestingly, the findings from this group showed that Piezo1 inhibition resulted in a sheet-like syncytium of MyHC coupled with increased fusion. Although these findings show contrasting results to the ones presented in this study, we must take into consideration potential factors which may explain why this may be the case. One such factor is the method of Piezo1 inhibition used by Tsuchiya et al. [32]. They carried out many of their experiments using knockout lines of Piezo1 through the gene editing tool CRISPR/Cas9. The fact that these cell lines did not express Piezo1 to begin with (unlike the cells we used) may yield completely different phenotypes, compared to the transient inhibition achieved by siRNA mediated transfection. Therefore, complete lack of Piezo1 expression may favour the activation of a secondary, yet unknown, alternative Piezo1-affected pathway(s) to fusion as observed for other factors involved in fusion, such as Myogenin [33]. Regarding the Piezo1 siRNA transfection experiments, although more than 60% reduction in gene expression is ideal; we nevertheless found that our level of Piezo1 knockdown produced very interesting effects on myogenic regulation. Moreover, Miyamoto et al. [22] also obtained slightly below or just about 60% Piezo1 reduction, and yet reported intriguing Piezo1-associated events in urothelial cells. Perhaps Tsuchiya et al., obtained even greater knockdown of Piezo1 in their siRNA-mediated analyses [32], further suggesting that the timing and level of Piezo1 expression may yield varying phenotypes. There was also the likelihood that siRNA used in this study may potentially have off-target effects on other genes which could influence myotube formation by employing other mechanistic pathways. However, this likelihood is reduced by the fact that we have now tested five different Piezo1-specific siRNAs (including the one used by Tsuchiya et al., from the company Sigma) and all show reduced fusion (Appendix A). Nevertheless, this study welcomes falsifiability. In fact, the siRNA used from Sigma, despite showing reduced fusion compared to controls, did show significantly greater fusion index compared to some of the siRNAs used from the company Qiagen. Furthermore, the knockdown efficiency measured by RT-qPCR showed that the Sigma-derived *Piezo1* siRNA was on average more efficient compared to the main siRNA from Qiagen used in this study (Appendix A). Thus, our report highlights the potential phenotypic differences from fine tuning Piezo1 downregulation levels. At the very least this report highlights the potential danger in using siRNAs solely from one provider and demands further high-quality testing from these companies to reduce potential off-target effects.

Additional support for the involvement of Piezo1 in myoblast fusion comes from our results which showed that its downregulation significantly reduced the expression of Myomaker–a muscle specific protein that localises to the plasma membrane and is crucial for vertebrate myoblast fusion [23,33,34,35,36]. Conversely Yoda1 specific activation of Piezo1 showed increased Myomaker expression (Appendix A) indicating an interplay between Piezo1 and Myomaker. Myomixer on the other hand did not show significant changes in expression post downregulation nor activation of Piezo1. This potential decoupling of these fusogenic proteins is in line with previous findings from Leikina et al. [36]. Future studies should increase the sample size and widen the time-course of analysis in order to uncover whether the respective decrease and increase in Myomaker expression is a direct response from the downregulation/activation of Piezo1 or an indirect event paralleling altered fusion pathway.

## 5. Conclusions

The data presented in this study showed that the Piezo1 channel is present in SC-derived myoblasts and myotubes but expressed at a higher proportion in the latter. Downregulation of Piezo1 significantly lowered the fusion capacity during myotube formation, growth and maturation. In contrast, Piezo1 activation increased fusion. Future research examining changes in myotube function (integrity, Ca^2+^ influx, cytoskeletal organisation, and fusion) that are directly the result of mechanical stress should consider analysis of Piezo1. In the context of therapeutic strategies against muscular dystrophies such as DMD, not only must we unravel the spatiotemporal regulation of Piezo1 expression, but we must be aware of this channel’s ability to alter its Ca^2+^ influx threshold by adapting or inactivating its gating capacity in response to repetitive stimuli. Pharmaceutically, small activating molecules, such as Yoda1, may prove beneficial. However careful attention must be given to the half-life and pharmacokinetics of these agonists in vivo before even considering them as viable drugs for human consumption. Piezo1′s importance in skeletal muscle maintenance and function will undoubtedly grow as new research aims to explore the mechanisms and signalling pathways this remarkable mechanosensor employs. Moreover, given Piezo1’s pivotal role in myogenic progression and its expression in satellite cells, it is likely that altered Piezo1 activity may associate with pathogenesis of satellite cell-opathies [37].

## Figures and Tables

**Figure 1 cells-11-00393-f001:**
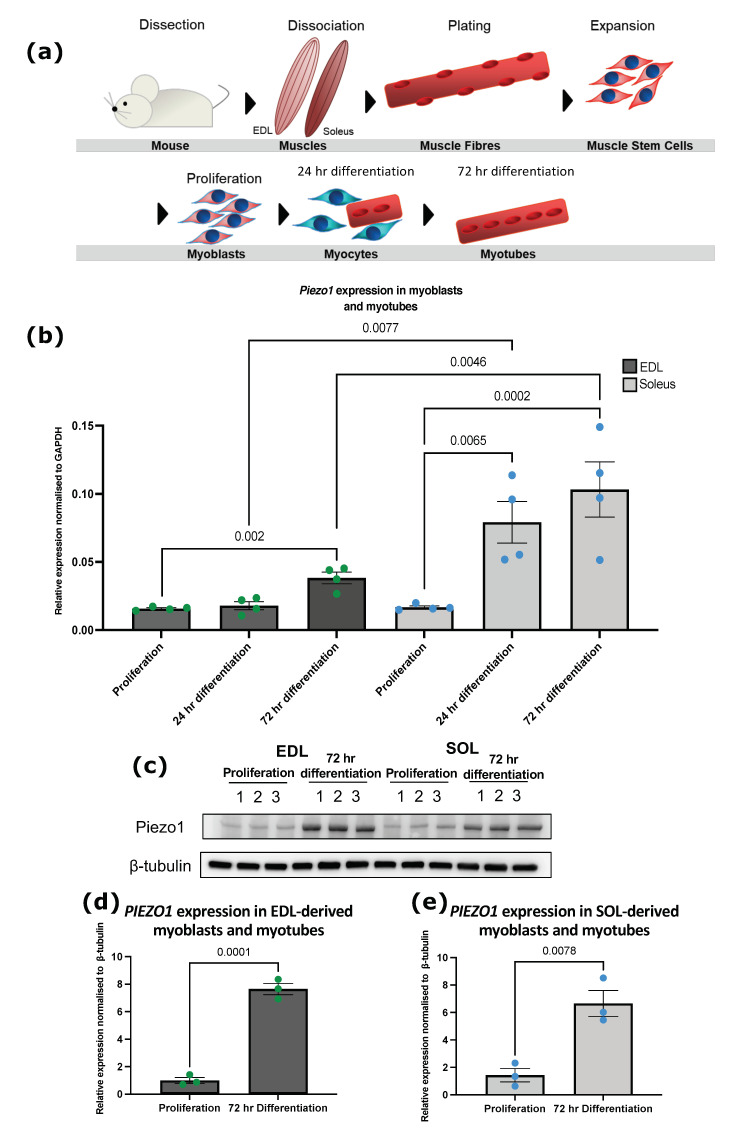
***Piezo1* expression increases in differentiating SC-derived myoblast.** (**a**) Schematic representation of muscle isolation and SC-derived myoblast expansion procedure. (**b**) Relative fold changes in expression of *Piezo1* in myoblasts from EDL (dark grey bars, green dots) and SOL muscle (light grey bars, blue dots), during proliferation and through differentiation; 24 h (Day 1) and 72 h (Day 3) in differentiation medium. Values were normalised to *Gapdh.* (**c**,**d**) Immunoblot analysis for Piezo1 protein in (**d**,**e**) EDL and I SOL-derived samples in proliferating and differentiated myotubes (72 h differentiation). Values were normalised to β-tubulin. Data are presented as mean ± SEM from four experiments (*n* = 3–4 mice). *p* values are annotated above graphs.

**Figure 2 cells-11-00393-f002:**
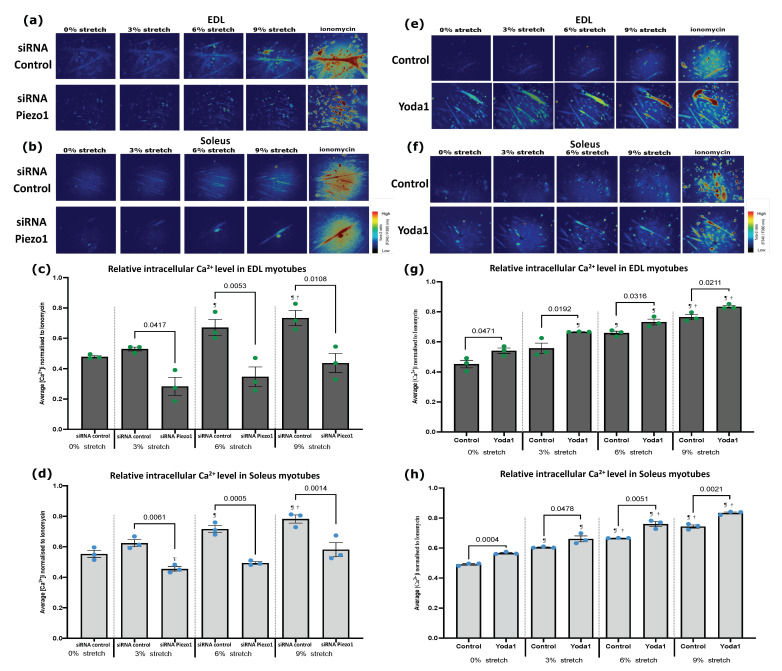
**Modulation of Piezo1 alters stretch-mediated increase of intracellular Ca^2+^ in myotubes.** (**a**,**b**) Representative images of intracellular Ca^2+^ imaging in EDL and SOL-derived myotubes. Myoblasts were transfected with 10 nM of either control-siRNA (top panels) or *Piezo1*-siRNA (bottom panels). After overnight incubation, cells were incubated for a further 72 h in differentiation medium. Fura 2-AM was administered to myotubes followed by a 30-min incubation time. Stretch was then applied at 3% (0.3 mm), 6% (0.6 mm) and 9% (0.9 mm) for 1 min followed by a 1-min resting period in between. Ionomycin at 5 µM was then applied. Side vertical bar shows Fura 2-AM ratio emittance from low to high. (**c**,**d**) Average changes in the intracellular Ca^2+^ level ([Ca^2+^]i) calculated by difference between base and peak pixel value normalised to ionomycin. (**e**,**f**) Representative images of Ca^2+^ imaging in EDL and SOL-derived myotubes. Fura 2-AM was administered to myotubes followed by a 30-min incubation time. Stretch was then applied at 3% (0.3 mm), 6% (0.6 mm), and 9% (0.9 mm) for 1 min followed by a 1-min resting period in between. During the initial 0% stretch timepoint, Yoda1 cohorts were administered with 30 µM of the agonist before being subjected to stretch (0% Yoda1 pictures were taken within 10–20 s after agonist administration). Ionomycin at 5 µM was then applied. Side vertical bar shows Fura 2-AM ratio emittance from low to high. (**g**,**h**) Average changes in the intracellular Ca^2+^ level ([Ca^2+^]i), difference between base and peak pixel value normalised to ionomycin. Data are mean ± SEM from three experiments (*n* = 3 mice). *p* values are annotated above graphs showing significance compared to control at each stretch condition. ¶: Significant difference at *p* < 0.05 compared to 0% stretch counterparts. †: Significant difference at *p* < 0.05 compared to 3% stretch counterparts. Ŧ: Significant difference at *p* < 0.05 compared to 9% stretch in *Piezo1*-siRNA conditions using one-way ANOVA followed by the Tukey-Kramer post-hoc.

**Figure 3 cells-11-00393-f003:**
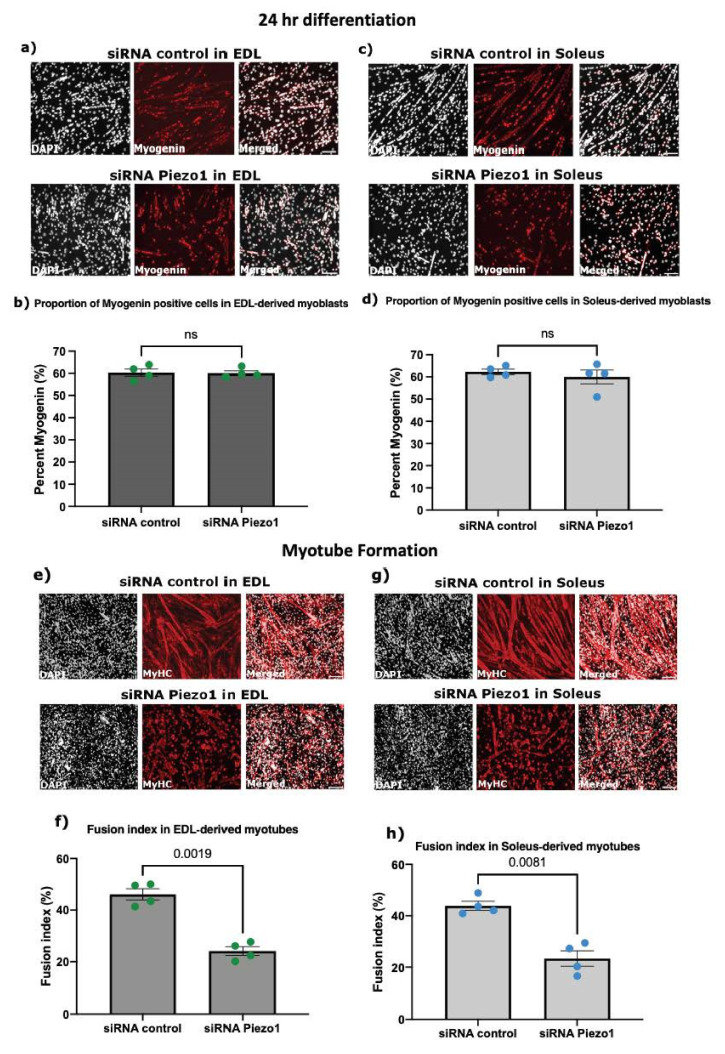
***Piezo1* knockdown reduces expression of fusogens and myoblast fusion.** (**a**,**c**) Representative images of EDL- and SOL-derived myoblasts transfected with 10 nM of control-siRNA (siRNA control) or *Piezo1*-siRNA. Following overnight incubation, cells were incubated for a further 24 h in differentiation medium, immunolabelled for Myogenin (red panels), and counterstained with DAPI (black and white panels). Scale bar is 100 μm. (**b**,**d**) Percentage proportion of Myogenin-positive cells relative to total nuclei. (*n* = 4 mice). (**e**,**g**) Representative images of myotubes from EDL and SOL, transfected with 10 nM of control-siRNA or *Piezo1*-siRNA. Following overnight incubation, cells were incubated for a further 72 h. Cells were immunolabelled for myosin heavy chain (MyHC) (red panels) and counterstained with DAPI (black and white panels). (**f**,**h**) The fusion index was calculated by counting the total number of nuclei within each myotube and representing this as a percentage relative to the total number nuclei in the image taken (*n* = 4 mice). (**i**–**l**) Relative fold changes in expression of EDL-derived myoblasts (**i**,**j**) and SOL-derived myoblasts (**k**,**l**). Cells were transfected with 10 nM of either control-siRNA (siRNA control) or *Piezo1*-siRNA (siRNA *Piezo1*). After overnight incubation, cells were incubated for a further 24 h. The expression of the fusogenic protein genes *Myomaker* and *Myomixer* were then analysed. Values were normalised to Gapdh. (*n* = 3 mice). (**m**,**o**) Representative images of EDL and SOL-derived myotubes. Early forming myotubes were transfected with 10 nM of control-siRNA (siRNA control) or *Piezo1*-siRNA (siRNA-*Piezo1*). Following overnight incubation, cells were incubated for a further 48 h, immunolabelled for myosin heavy chain (MyHC) (red panels) and counterstained with DAPI (black and white panels). Scale bar is 100 μm. (**n**,**p**) Bar graphs display the fusion index (*n* = 3 mice). Data are mean ± SEM. *p* values are annotated above graphs showing significance (or ns, not significant) compared to siRNA control conditions using a two-tailed paired Student’s *t*-test.

**Figure 4 cells-11-00393-f004:**
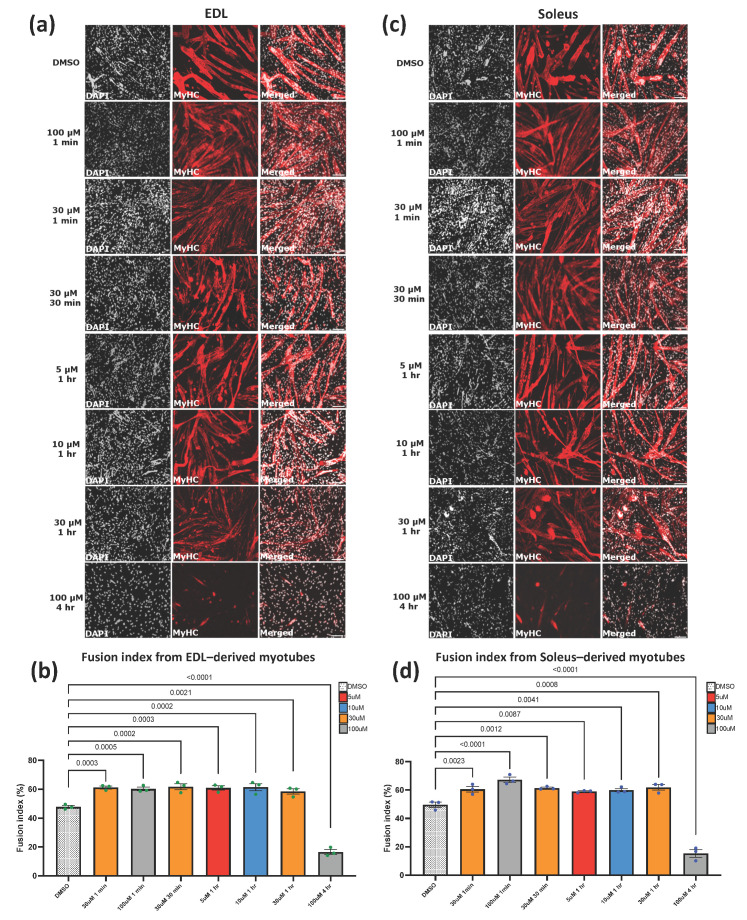
**Piezo1 activation increases myoblast fusion at the expense of myotube syncytial maturation.** (**a**,**c**) Representative images of cohorts at relevant timepoints and concentrations, immunostained for myosin heavy (MyHC) (red panels) and counterstained with DAPI (black and white panels). Micrographs taken at ×20 magnification. Scale bar is 100 µm. (**b**,**d**) Bar graphs displaying fusion index in EDL- and SOL-derived myotubes at the relevant time point and concentration variables. Yoda1 values are compared to DMSO controls at 1 min. Please refer to Appendix A for cross comparison of all concentrations against their respective controls at each timepoint. (**e**,**g**) Representative images of EDL- and SOL-derived myotubes immunolabelled for MyHC. We measured the width (diameter) of the myotubes to quantify potential differences between Piezo1-activated cohorts (lower panels) and DMSO controls (upper panels). This was achieved by measuring all myotube width in each field of view and taking the top five widest diameters from each (representative yellow bars). Example pictures from DMSO controls and 30 µM of Yoda1 incubated for 30 min are displayed. Scale bar is 100 µm. (**f**,**h**) Average myotube width in DMSO and Piezo1 activated samples. (**i**,**j**) Proportion of myotube width in EDL- and SOL-derived myotubes. Myotube width was divided into incremental bins of 5 µm and represented as percentages relative to the total number of myotubes counted. Values are mean ± SEM. *p* values are annotated above graphs showing significance compared to DMSO controls using one-way ANOVA tests followed by the Tukey-Kramer post-hoc. *n* = 3 mice. For (**f**,**h**) asterisks (*) denote significance at *p* < 0.0001 compared to DMSO controls conditions. Please note that only cohorts which showed increased myoblast fusion in both EDL- and SOL-derived myotubes from Figure 4a–d are displayed.

**Table 1 cells-11-00393-t001:** List of siRNAs used.

Gene	Species	siRNA ID
***scrambled non-targeting siRNA*** (All Stars Negative Control siRNA)	Mouse	Qiagen, 1027281
** *Piezo1* **	**Mouse**	**Qiagen, S104420409**
*Piezo1*	Mouse	Qiagen, S104420402
*Piezo1*	Mouse	Qiagen, S100814807
*Piezo1*	Mouse	Qiagen, S100814821

N.B. The *Piezo1* siRNA in bold (S1044120409) was used for most of the experiments. The other three were used as validators of our obtained results.

**Table 2 cells-11-00393-t002:** RT-qPCR primers.

Gene	Species	Forward Primer (5′-3′)	Reverse Primer (5′-3′)	Reference
** *Gapdh* **	Mouse	TGTGTCCGTGGATCTGA	TTGCTGTTGAAGTCGCAGGAG	Takara Bio, MA050371
** *Piezo1* **	Mouse	CTTTATCATGAAGTGCAGCCGAG	CCAGATGATGGCGATGAGGA	Takara Bio, MA125411
** *Myomaker* **	Mouse	CATGCGCCGTGACATTCTG	AAGCATTGTGAAGGTCGATCTCTG	Takara Bio, MA131293
** *Myomixer* **	Mouse	GAATCCACCGCAGGCAAA	ACCATCGGGAGCAATGGAAC	Takara Bio, MA101853

**Table 3 cells-11-00393-t003:** Primary and Secondary antibodies used.

Primary Antibody	Dilution	Reference
**Monoclonal mouse**—Myogenin	1:10	Development Studies Hybridoma Bank (DSHB), F5D-s
**Monoclonal mouse—MF20 (myosin heavy chain)**	1:10	Development Studies Hybridoma Bank (DSHB), MF20-s
**Polyclonal rabbit—Piezo1**	1:100	Proteintech, 15939-1-AP
**Secondary antibody**	**Dilution**	**Reference**
**Donkey anti-mouse IgG (H + L), Alexa Fluor^®^ 555**	1:500	Life Technologies, A21203
**Donkey anti-rabbit IgG (H + L), Alexa Fluor^®^ 488**	1:500	Life Technologies, A-21206

## Data Availability

All data are available upon request.

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
