# Peer review of "Fine-Tuning of Piezo1 Expression and Activity Ensures Efficient Myoblast Fusion during Skeletal Myogenesis"

_cells, 2022, doi:10.3390/cells11030393_

Round 1

Reviewer 1 Report

The study by Huascar Pedro Ortuste Quiroga et al. investigated the role of Piezo1 channels in myoblast differentiation into myotubes in vitro. Myoblast stem cells isolated from EDL and soleus were cultured and the expression (qPCR) and functional activity (stretch activation-induced Ca2+ influx) were assessed at different stages (i.e., proliferation vs. different differentiation times). The authors showed that Piezo1 expression increases during myoblast differentiation and, using siRNA techniques, showed blunted expression of Piezo1 resulted in reduced stretch-induced Ca2+ influx, myotube formation and fusion, and the expression of differentiation fusogens in differentiated myotubes mostly independent of muscle type. In contrast, the authors showed that over-activation, using different Yoda1 concentrations over time, of Piezo1 results in dysregulated tube formation. While this is an interesting study that shows the importance of Piezo1 in tube formation that very likely depends on Piezo-mediated Ca2+ influx, there are a few major concerns with the aspects of the data analysis and interpretation, as well as clarity of methodology that need to be addressed. Please see comments below:

Major

1. Some of the authors claims are made based on studies with low n and large variance (Supplemental Fig3j,l) or have low n but are trending (e.g., Fig3g,l).

2. Figure 2 - it is unclear what the Yoda + stretch activation data contribute, especially without a 0% stretch + Yoda in the control group. Can the authors compare deltas across the different stretch %s to perhaps show that as more Piezo1 is activated with increased stretch, less is available for activation by Yoda1? Similarly, can the authors increase stretching (12-15%) to show that Yoda1 no longer promotes an increase, or a significantly blunted increase, in Ca2+ influx. Representative images of ionomycin-induced Ca2+ influx would be beneficial.

3.  It is clear that Piezo1 contributes to stretch-induced Ca2+ influx, but the expression data regarding Piezo2 does not mean that Piezo2 cannot contribute. Knockdown experiments should be repeated for Piezo2. Though the expression of Piezo1 appears to be an order of magnitude greater than Piezo2, Piezo1 expression also appears low.

4. Figure 4 – The method of assessing tube width needs additional clarity. The authors indicate that measurements were taken at the widest distance in myotube regions with branches but this is not clear in the representative images (e.g., Fig4g bottom panel).

5. Supplemental Fig4 - The authors infer that Piezo1 likely indirectly affects cytoskeletal organization through evidence that Piezo1 knockdown prevents f-actin accumulation and Piezo1 overactivation has no effect on or decreased (when excessive) f-actin accumulation. More mechanistic evidence is required to show how Piezo1 activity affects (directly or indirectly) cytoskeletal dynamics.

Minor

  1. Figures should be described in the appropriate order in the text.
  2. There is no mention of Supplemental Fig5 in the Results, but it is discussed in the Discussion as a larger issue with siRNAs in the literature. While this type of discussion should perhaps be addressed, the data presented are out of place for this study.

Reviewer 2 Report

The current manuscript titled “Fine tuning of Piezo1 Expression and Activity Ensures Efficient Myoblast Fusion during Skeletal Myogenesis” described the role of Piezo1 channel in myoblast fusion. Authors found the increased Piezo1 expression in differentiating SC-derived myoblast and Piezo1 knockdown suppressed myotube formation and calcium influx induced by stretch. The topic of this MS is interesting and not many references was found in PubMed. I have some concerns for this MS before it is suitable to be published in the journal. 

  1. In figure1, authors used RT-PCR to detected the RNA of Piezo1 in differentiating SC-derived myoblast. This is a key experiment for the whole hypothesis. The level of Piezo2 may also need to be detected and only RT-PCR evidence is poor to indicate the increased expression of Piezo1. Western blots may be a good candidate.
  2. In the figure 2e, stretch-indued calcium infux is not clear in e (control). To confirm the increased calcium signal is mediated by Piezo 1 , the calcium signal shoule be detected in the calcium free ex-solution. I did not get the sense of Yoda1 application. On the other hand, if Yoda1 can rescue the calcium influx in present of siRNA for Piezo 1, that will be support for the conclusion.
  3. As the authors stated that Piezo1 deregulation might have implications in muscle aging and degenerative diseases including muscular dystrophies. If any available animal models or cell model for muscle aging or muscular dystrophies, it will be useful to test the expression of the channel and provide stronger support for this conclusion.  

Round 2

Reviewer 1 Report

The authors adequately addressed only some of the concerns from the original manuscript. There are existing issues.

  1. It is not sufficient to acknowledge the need for increased n in future studies. It is recommended that these preliminary studies be removed to be expanded on further in a future study or n to be increased for inclusion in the present study.
  2. There are still existing issues with Figure 4. First, simply moving the yellow bars does not improve the clarity of the methodology. Second, and more importantly, moving the yellow bars did not appear to alter the group data and it certainly should have. Please check the group data. Lastly, the methods text still needs improving. Were these analyses conducted by a blinded investigator? How were the independent images chosen? The way this is described in the text makes it appear as though investigator bias could have easily been introduced.

Reviewer 2 Report

My major concerns have been addressed.  

Round 3

Reviewer 1 Report

The authors have addressed all concerns.